# *Helicobacter pylori cagA*, *vacA*, *iceA* and *babA* Genotypes from Peruvian Patients with Gastric Intestinal Metaplasia

**DOI:** 10.3390/cancers16081476

**Published:** 2024-04-12

**Authors:** Jesús Guzmán, Denis Castillo, Anabel D. González-Siccha, Alejandro Bussalleu, Alba A. Trespalacios-Rangel, Andres G. Lescano, Michel Sauvain

**Affiliations:** 1Laboratorio Centinela de Helicobacter pylori, Instituto de Medicina Tropical Alexander von Humboldt, Universidad Peruana Cayetano Heredia, Lima 15024, Peru; denis.castillo.p@upch.pe (D.C.); alejandro.bussalleu@upch.pe (A.B.); michel.sauvain@ird.fr (M.S.); 2Facultad de Salud Pública y Administración, Universidad Peruana Cayetano Heredia, Lima 15102, Peru; andres.lescano.g@upch.pe; 3Departamento de Bioquímica, Facultad de Farmacia y Bioquímica, Universidad Nacional de Trujillo, Trujillo 13011, Peru; agonzalez@unitru.edu.pe; 4Grupo de Investigación en Enfermedades Infecciosas, Departamento de Microbiología, Facultad de Ciencias, Pontificia Universidad Javeriana, Bogotá 110231, Colombia; alba.trespalacios@javeriana.edu.co; 5UMR 152 Pharmacochimie et Biologie pour le Développement (PHARMA-DEV), Institut de Recherche pour le Développement (IRD), Université de Toulouse, CEDEX 9, 31062 Toulouse, France

**Keywords:** *Helicobacter pylori*, genotyping, virulence factor, *cagA*, *vacA*, *iceA*, *babA*, intestinal metaplasia, Peru

## Abstract

**Simple Summary:**

Virulence factor genes in *Helicobacter pylori* (*H. pylori*) strains promote changes in the gastric epithelial mucosa that are associated with the risk of developing neoplastic lesions, and in Peru, *H. pylori* infection prevalence is greater than 45% and the gastric cancer incidence rate is the highest in the Americas. The aim of our cross-sectional study was to explore the association between clinical strain virulence genotypes of *H. pylori* from Peruvian patients with gastric intestinal metaplasia compared to chronic non-atrophic gastritis cases. We observed that the prevalence of the genotypes *cagA+/vacAs1m1* and *cagA+/vacAs1am1* was 2.42 and 1.67 times higher in cases with intestinal metaplasia compared to chronic non-atrophic gastritis cases, respectively. Our findings revealed that *H. pylori* strains circulating in our environment have a higher frequency of genotypes documented as risk variants for neoplastic lesions, highlighting that in Peruvian patients with *H. pylori* infection, the risk genotypes are related to intestinal metaplasia clinical stage.

**Abstract:**

We explored the clinical-stage association of gastric intestinal metaplasia (IM) compared to cases of chronic non-atrophic gastritis (CNAG) and its relationship with virulence genotypes of *Helicobacter pylori* (*H. pylori*) clinical isolates from patients with dyspepsia in Peru. This study was cross-sectional and included 158 *H. pylori* clinical isolates; each isolate corresponded to a different Peruvian patient, genotyped by polymerase chain reaction to detect *cagA* gene and EPIYA motifs, the *vacA* gene (alleles *s1*, *s2*, *i1*, *i2*, *d1*, *d2*, *m1*, *m2* and subtypes *s1a*, *s1b* and *s1c*), the *iceA* gene (alleles *1* and *2*), and the *babA* gene (allele *2*). We observed that 38.6% presented with IM and that all clinical isolates were *CagA* positive. The EPIYA-ABC motif was predominant (68.4%), and we observed a high frequency for the *vacA* gene alleles *s1* (94.9%), *m1* (81.7%), *i1* (63.9%), and *d1* (70.9%). Strains with both *iceA* alleles were also detected (69.6%) and 52.2% were *babA2* positive. In addition, it was observed that the *cagA+/vacAs1m1* (PR: 2.42, 1.14 to 5.13, *p* < 0.05) and *cagA+/vacAs1am1* (PR: 1.67, 1.13 to 2.45, *p* < 0.01) genotypes were associated with IM. Our findings revealed the *cagA* and *vacA* risk genotypes predominance, and we provided clinically relevant associations between Peruvian patients with *H. pylori* infection and IM clinical stage.

## 1. Introduction

Gastric cancer (GC) is the sixth most common cause of death due to malignancy worldwide, and *Helicobacter pylori* (*H. pylori)* infection is the GC leading cause among infection-related cancer diagnoses. In the Americas, Peru has the GC highest incidence for both sexes (age standardized rate per-100,000 people: 10.9), followed by Colombia (10.2), Chile (9.8) and Haiti (9.7) [1,2,3]. GC in Peru represents the second leading cause of death by malignancy [4], *H. pylori* infection prevalence ranges from 45% to 83%, and it is estimated that 60.8% of patients with a GC diagnosis are infected by *H. pylori* [5,6].

Gastric adenocarcinoma development is initiated by a series of changes in the gastric epithelial mucosa with highly differentiated patterns, which is described as the Correa Cascade [7,8], and it is known that persistent infection and the presence of genes related to virulence factors in *H. pylori* are associated with the risk of developing preneoplastic lesions. However, not all patients with chronic infection develop severe lesions on the gastric epithelium [9,10].

Genetic variability in genes encoding *H. pylori* virulence factors may modulate risk for the development of malignant lesions [11]. Gene-A-associated cytotoxin is a protein encoded by the *cagA* gene and can translocate by means of a type IV secretion system and induce a pro-inflammatory response in gastric epithelial cells that may promote precancerous and cancerous lesions. Vacuolating cytotoxin A gene encodes the vacA cytotoxin and can induce an acute inflammatory response in the gastric epithelium, generating important promoter activity for the development of neoplastic lesions [12,13]. In addition, virulence factors linked to *H. pylori* colonization and adhesion on the gastric epithelial mucosa, such as the epithelial-contact-induced gene (*iceA*) and the blood group antigen-binding adhesin gene (*babA*), are associated with the presence of ulcerative or atrophic gastric lesions, promoting the risk of preneoplastic lesions [14,15,16].

Each geographic region has its own profile of genetic variants based on the distribution of virulence factors among its circulating strains, and in Peru, a variable frequency of *H. pylori* strains positive for the *cagA* gene (79.9% to 100%) and in the detection of genotypes *s1* (41.6% to 100%) and *m1* (60.7% to 75%) genotypes of the *vacA* gene has been reported [17,18], In addition, a frequency of 33.3% for the *vacAs1m1* genotype was found in cases with a diagnosis of GC and *H. pylori* infection [6].

Despite the high incidence of CG and the high frequency of *H. pylori* infection in Peru, information on the virulence genotypes of *H. pylori* strains is still scarce and little is known about their relationship with the clinical stage of Peruvian patients. Therefore, the aim of this study was to determine the association between genes associated with *H. pylori* virulence factors [*cagA* status, EPIYA motifs, *vacA* gene polymorphisms (*s*, *m*, *i* and *d)*, *iceA* gene alleles 1 and 2, and *babA* gene allele 2] and the clinical stage of preneoplastic lesion gastric intestinal metaplasia observed in a sample of Peruvian patients.

## 2. Materials and Methods

### 2.1. Study Population and Design

This study was descriptive, cross-sectional, and included 158 *H. pylori* clinical isolates from adult outpatients with dyspepsia referred for endoscopy at the gastroenterology service of a university hospital in Lima, Peru, and diagnosed with *H. pylori* infection for the first time (between March 2016 and August 2017). The strains were obtained from a previous study conducted by researchers at the *Helicobacter pylori* Sentinel Laboratory at the Alexander von Humboldt Institute of Tropical Medicine in Lima, Peru. The protocol was approved by the Institutional Human Ethics Committee of the Universidad Peruana Cayetano Heredia (File 059-03-20 with registration code 221182). Strains were isolated from gastric biopsies obtained according to the Maastricht protocol [19], and each isolate was from a different patient.

Of the patients from whom the strains were isolated, the histopathological report was obtained, and the cases were classified into clinical stages of gastric intestinal metaplasia (IM) (N = 61, 38.6%) and chronic non-atrophic gastritis (CNAG) (N = 97, 61.4%). Histopathological analysis was performed using the Sydney system by hematoxylin–eosin staining of gastric samples [20]. The sample size was estimated by power calculation based on a pilot trial with a subsample of the analyzed cohort, where a prevalence of cases with IM-presenting isolates with the *cagA/vacAs1am1* genotype of 58.3% and a prevalence of 34.5% of cases with other IM-presenting genotypes were considered. The POWER command in Stata/SE version 17.0 software was used to determine the statistical power, resulting in a power of 84.3%.

### 2.2. H. pylori Culture Conditions

Clinical isolates were reactivated independently, and a 200 µL aliquot of the homogenate was plated on BHI agar plates (BD, Heidelberg, Germany) supplemented with 10% defibrinated lamb blood, amphotericin B (Sigma-Aldrich, St. Louis, MO, USA), and Skirrow (Oxoid, Basingstoke, UK). Culture plates were incubated under microaerobiosis conditions (10% CO_2_, 5% O_2_, and 85% N_2_) at 37 °C for 4 to 7 days. DNA from each *H. pylori* strain was extracted according to the GeneJET Genomic DNA Purification Kit protocol (ThermoFisher, Lenexa, KS, USA) [21]. The purified DNA was quantified by Nanodrop (ThermoFisher, Lenexa, KS, USA), its integrity was evaluated by 0.6% agarose gel electrophoresis, and it was stored at 4 °C [22].

### 2.3. PCR Amplification and Typing

Molecular typing was performed by conventional PCR and yielded amplification products for the *cagA* gene, which included the detection of the EPIYA-A, EPIYA-B and EPIYA-C motifs, the *vacA* gene (alleles *s1*, *s2*, *i1*, *i2*, *d1*, *d2*, *m1*, *m2* and subtypes *s1a*, *s1b* and *s1c*), the *iceA* gene (alleles *1* and *2*), and the *babA* gene (allele *2*) [23,24,25,26,27,28,29,30,31,32,33]. The genes were typed using the enzyme Taq DNA Polymerase GoTaq Green Master Mix (Promega, Madison, WI, USA), with final reaction volumes of 20 μL and 10 μM of the primers (Table 1). Amplification products were visualized using 2% agarose gel electrophoresis. *H. pylori* reference strain ATCC 43504 was used as control.

### 2.4. Statistical Analysis

The association between the preneoplastic lesion types of the gastric epithelial mucosa and the genotype of *H. pylori* isolates was evaluated through a bivariate analysis using the chi-squared test, and in cases of violation of assumptions, Fisher’s exact test was used. A binomial-log generalized linear regression analysis was performed, and the magnitude of the association was expressed as a prevalence ratio (PR) as an estimate relative to the risk with a 95% confidence interval (95% CI) regression. A statistical significance level was considered for values of *p* < 0.05. All analyses were performed using the statistical package Stata/SE version 17.0.

## 3. Results

### 3.1. Patient Demographics

In the population of 158 cases of dyspepsia with *H. pylori* infection, a higher frequency of infection was observed in the female sex (70.3%) and in the age group over 50 years (58.8%). In addition, 38.6% of cases were histopathologically diagnosed as IM, with a higher frequency of IM in the age group over 50 years (*p* < 0.05) (Table 2).

### 3.2. cagA Genotypes and EPIYA Motifs

All isolates were positive for the *cagA* gene, and the EPIYA motifs of the cagA protein corresponded to the EPIYA-A, EPIYA-B, and EPIYA-C segments, where there were single and multiple *cagA*-EPIYA genotypes and a predominance for the *cagA-ABC* genotype. It was observed that 114 cases had no co-infection (6 EPIYA-AB cases (3.8%); 108 EPIYA-ABC cases (68.4%)). In addition, among the cases with co-infection, 33 cases were observed with a dual EPIYA genotype with AB/ABC (3.2%) and ABC/ABCC (17.7%) profiles and 11 cases with a triple EPIYA genotype (ABC/ABCC/ABCCC) (Table 3).

### 3.3. vacA Genotypes

A high frequency was observed for the *vacAs1* allele (94.9%) and the *vacAs1a* subtype (48.1%) of the signal region of the gene. The findings in the middle region, intermediate region, and deletion of the intermediate region of the *vacA* gene showed a high frequency for the genotypes *vacAm1* (81.7%), *vacAi1* (63.9%), and *vacAd1* (70.9%). It was also observed that the *vacAs1* subtypes (*vacAs1a* genotype: 43.4%), the middle region (*vacAm1* genotype: 43.4%), and the intermediate region (*vacAi1* genotype: 47.5%) were statistically significant regarding IM diagnosis. In addition, the subtyping of the *vacA* gene signal region revealed co-infection with multiple genotypes (*s1a/s1b*, *s1a/s1c*, and *s1a/s1b/s1c*) (Table 3).

### 3.4. iceA and babA2 Genotypes

*H. pylori* strains presented both alleles for the *iceA* gene in 69.6% of cases, and 52.5% of isolates were positive for the *babA2* gene. In addition, *H. pylori* strains were positive for allele *1* of the *iceA* gene in 51.5% of cases with IM, which was statistically significant (Table 3).

### 3.5. Relation of Virulence Genotypes with IM

The EPIYA-*ABC* motif was predominant, and EPIYA-C segment duplication was statistically associated with and prevalent in IM with respect to the detection of a single EPIYA-C segment (prevalence ratio (PR): 1.56, 95% confidence interval (95% CI): 1.02 to 2.41, *p* < 0.05). In relation to the *vacA* gene, *m1* and *i1* alleles had significant associations with IM with respect to *m2* and *i2* polymorphisms (PR: 2.52, 95% CI: 1.11 to 5.74, *p* < 0.05 for *m1*; PR: 2.32, 95% CI: 1.25 to 4.32, *p* < 0.01 for *i1*). Clinical isolates carrying both alleles (allele *1* and allele *2*) of the *iceA* gene were also observed and were significantly associated with a lower frequency of IM cases than *iceA2* clinical isolates (PR: 0.46, 95% CI: 0.29 to 0.73, *p* < 0.01). The allele 2 of the *babA* gene did not show a significant association with clinical-stage IM. In addition, it was observed that the co-detection of both the positive status of the *cagA* gene and the presence of *s1*, *s1a*, and *m1* polymorphisms of the *vacA* gene were statistically associated with and prevalent in IM with respect to other documented genotypes with lower virulence (*cagA+/vacAs1m1*, PR: 2.42, 95% CI: 1.14 to 5.13, *p* < 0.05) (*cagA+/vacAs1am1*, PR: 1.67, 95% CI: 1.13 to 2.45, *p* < 0.01) (Table 3).

## 4. Discussion

IM can be considered a point of no return in the GC cascade, and it is believed that patients with IM are at high risk for GC even after *H. pylori* eradication [34,35]. We observed that preneoplastic condition IM presented a significant association with age and that its prevalence was 2.4 times higher in patients older than 50 years compared to cases younger than 35 years (95% CI: 0.97 to 5.88, *p* = 0.05). This finding is similar to those regarding the age-related preneoplastic lesions reported in patients infected with *H. pylori* [36,37,38] and may be a chronic effect of the infection due to cumulative damage to the epithelial mucosa.

Genetic and epigenetic changes on gastric cells are induced by gene coding for virulence factor genes in *H. pylori* [39]. *CagA*-positive *H. pylori* strains are known to promote toxicity mechanisms that increase the risk of GC [23,40,41,42]. In Peru, high frequencies of strains positive for the *cagA* gene have been reported (79.9% to 100%) [6,17,18], and we confirm these findings. In our study, we observed that all clinical isolates were positive for the *cagA* gene, suggesting a predominance of the *cagA*-positive genotype among circulating *H. pylori* strains and predisposing infected cases to a higher degree of chronic inflammation with a risk of preneoplastic gastric lesions [43].

We also characterized the EPIYA segments of the C-terminal region of the *cagA* gene; as in many reports [44,45,46,47], we observed that an increase in EPIYA-C segment repeats is a risk factor for preneoplastic gastric lesions. The analysis showed that the IM condition was prevalent and associated with the presence of *H. pylori* strains with two EPIYA-C segments (PR: 1.56, 95% CI: 1.02 to 2.41, *p* < 0.05), which may be related to the pro-oncogenic effects of the EPIYA-C segment on gastric cells [48]. In this study, *H. pylori* strains with three EPIYA-C segments were detected, and their frequency was lower than detections with two EPIYA-C segments (6.9% and 17.7%, respectively), but the association with IM condition was absent. In addition, we observed multiple *cagA* EPIYA genotypes in 27.8% of cases, suggesting the presence of different *H. pylori* strains from the same patient or the presence of new strains arising from the high recombination frequency and microevolution of *H. pylori* [24], which could confound the potential effect of the virulence marker and lead to a Type II error [44].

Vacuolating cytotoxin gen *(vacA*) is an important virulence marker in *H. pylori* and is known to induce vacuolization of gastric epithelial cells in a variable manner due to its ability to assume different polymorphic rearrangements due to the genetic diversity of the *vacA* gene [25]. We confirm the findings reported in Peru [17,18], and we observed a predominance of *H. pylori* strains with *vacAs1*, *vacAm1*, and *vacAi1* polymorphisms. The analysis showed no association of the *vacAs1* polymorphism with the IM condition, but in similarity to what was reported in clinical isolates from Colombia. [49], the frequency of the *vacAs1* polymorphism and its *vacAs1a* variant was predominant. In addition, we observed that when characterizing the *vacAs1* polymorphism, 51.9% of the cases presented multiple genotypes and that co-infection by *vacAs1a* and *vacAs1b* strains was inversely associated with the IM condition (PR: 0.40, 95% CI: 0.18 to 0.88, *p* < 0.05). It has been suggested that the presence of co-infection could reduce the potential risk of a virulence factor causing preneoplastic injury, and therefore warrants a thorough evaluation [50].

We also observed that the *vacAm1* polymorphism (PR: 2.52, 95% CI: 1.11 to 5.74, *p* < 0.05) is a significant risk factor for IM. Our findings are similar to the meta-analysis published by Sugimoto et al., which compared studies of populations from Mexico, Costa Rica, Colombia, Brazil, Venezuela, Chile, and Argentina and reported that the *vacAm1* polymorphism (OR: 3.59, 95% CI: 2.27 to 5.68, *p* < 0.05) is a risk factor for neoplastic lesions (OR: 3.59, 95% CI: 2.27 to 5.68, *p* < 0.05) [51]. We also observed that *vacAi1* polymorphism (PR: 2.32, 95% CI: 1.25 to 4.32, *p* < 0.05) was associated with the IM condition; these findings are similar to reports of patients from Morocco and Iran, where it is observed to be a risk factor for IM (OR: 2.75, 95% CI: 1.59. to 4.73, *p* < 0.05), and the clinical condition of chronic atrophic gastritis (OR: 2.8, 95% CI: 1.45. to 5.40, *p* < 0.05) [52,53].

In our study, we also reported that the *vacAd1* polymorphism was predominant but was not associated with the IM condition. According to Ogiwari et al., the *vacAd1* polymorphism could be a predictor of gastric neoplastic lesions and reports frequencies in populations from the United States (74.1%) and Colombia (74%) that are comparable to our study. Furthermore, it reports significant risks for cases with GC (OR: 8.04; 95% CI: 2.67 to 24.16; *p* < 0.05) but no association in patients with preneoplastic lesions (OR: 1.87; 95% CI: 0.91 to 3.87; *p* > 0.05) [31].

We have studied the allelic variants of the *iceA* gene, and although its function is still uncertain, it is known that its expression is induced by contact between *H. pylori* and gastric cells [54]. Our study showed that the frequency of *iceA1* was higher than that of *iceA2* (20.9% and 9.5%, respectively), and that both polymorphisms were prevalent among IM cases (51.5% for *iceA1* and 66.7% for *iceA2*). This finding differs completely from that reported for Western countries [14,16], but shows similarity to the distribution of *iceA1* strains [55] and with the frequencies observed among cases with neoplastic lesions in Colombia (52.1% for *iceA1* and 69.5% for *iceA2*) [49]. This suggests that the allelic distribution of *iceA* may be related to the geographical distribution of *H. pylori* strains [16]. Likewise, several studies have reported co-infection by *iceA1* and *iceA2* strains [14,16,56,57]; co-infection was frequent (69.6%) in our study and prevailed in the clinical condition CNAG (69.1%). Moreover, it was significantly inversely associated with IM (PR: 0.46; 95% CI: 0.29 to 0.73, *p* < 0.05).

On the other hand, we characterized allele *2* of the *babA* gene and observed a frequency of 52.5% of *babA2* strains, which is comparable to the frequencies observed (40.4% to 82.3%) in populations of the Americas [55,58,59,60]. Adhesin babA is a determinant in *H. pylori* gastric colonization [15,61] and it has been reported that the presence of the *babA2* gene is related to GC [33,62]; however, it is suggested that the risk relationship with neoplastic lesions may be especially associated with Asian populations but not with South American populations [63], which is similar to our finding (PR: 1.13; 95% CI: 0.76 to 1.69, *p* > 0.05).

In addition, the risk of IM or GC is known to be increased by infection with *H. pylori* strains that simultaneously display multiple virulence genes [49,64]. We observed that both the *cagA+/vacAs1m1* genotype (PR: 2.42, 95% CI: 1.14 to 5.13, *p* < 0.05) and the *cagA+/vacAs1am1* genotype (PR: 1.67, 95% CI: 1.13 to 2.45, *p* < 0.01) were prevalent in the IM condition. It has been reported that the risk of gastric neoplastic lesions in cases with *vacAs1m1* genotype infection increases from 1.75 (95% CI: 1.04 to 2.96)-fold to 4.8 (95% CI: 1.71 to 13.5)-fold given the detection of *a cagA+* gene-positive status. [12,65]. In our study, the simultaneous detection of virulence polymorphisms for the *cagA* and *vacA* genes was associated with the IM condition; however, it is suggested that the strength of association might have been underestimated given the absence of *cagA*-negative *H. pylori* genotypes among our circulating strains; so, these findings should be taken with caution and should be considered significant.

Finally, our study was subject to limitations inherent to its cross-sectional nature and its ability to differentiate cause-and-effect relationships in the genotype–phenotype association. Furthermore, the sample size, the absence of healthy cases and the absence of cases with clinical stages such as dysplasia or gastric adenocarcinoma prevent the generalization of the results. However, in our study, *H. pylori* characterization has important clinical significance, but its relationship to the preneoplastic clinical stage may not reveal its potential risk given the influence of other variables such as the genetic variability of *H. pylori* or the presence of coinfection, the genetic susceptibility of the host, the responsiveness of the host immune system, and environmental factors [66,67].

## 5. Conclusions

Our findings revealed that the *H. pylori* strains circulating in our environment present a higher frequency of genotypes documented as risk variants for neoplastic lesions, and that their distribution was observed both for cases with CNAG and for cases with IM, highlighting that in Peruvian patients with dyspepsia who present *H. pylori* infection, the presence of risk polymorphisms for the *cagA* and *vacA* genes is related to the clinical stage of preneoplastic IM lesion.

## Figures and Tables

**Table 1 cancers-16-01476-t001:** Protocols used for genotyping *H. pylori* clinical isolates.

Gene	Region or Allele	Primers (5′–3′)	Amplified Product (bp)	PCR Protocol
*cagA*	Constant 5′	F: TTGACC AACAACCACAAACCGAAG R: CTTCCCTTAATTGCGAGATTCC	183	1 cycle of 95 °C for 9 min, 40 cycles of 95 °C for 30 s, 50 °C for 45 s and 72 °C for 45 s, and 1 cycle of 72 °C for 5 min [23]
	Variable 3′	F: ACCCTAGTCGGTAATGGG R: GCTTTAGCTTCTGAYACYGC	400 (AB) 500 (ABC) 600 (ABCC) 700 (ABCCC)	1 cycle of 95 °C for 10 min, 39 cycles of 95 °C for 30 s, 52.3 °C for 30 s and 72 °C for 36 s, and 1 cycle of 72 °C for 5 min [24]
*vacA*	Signal	F: ATGGAAATACAACAAACACAC R: CTGCTTGAATGCGCCAAAC	259 (s1) 286 (s2)	1 cycle of 95 °C for 2 min, 40 cycles of 95 °C for 30 s, 52 °C for 30 s and 72 °C for 30 s, and 1 cycle of 72 °C for 5 min [25,26]
	Signal (sub)	Fs1a: GTCAGCATCACACCGCAAC Fs1b: AGCGCCATACCGCAAGAG Fs1c: TTAGTTTCTCTCGCTTTAGTRGGGYT R: CTGCTTGAATGCGCCAAAC	190 (s1a) 187 (s1b) 220 (s1c)	1 cycle of 95 °C for 2 min, 35 cycles of 95 °C for 60 s, 52 °C for 60 s and 72 °C for 60 s, and 1 cycle of 72 °C for 5 min [25,26]
	Middle	F: CCATCTGTCCAATCAAGCGAG R: GCGTCTAAATAATTCCAAGG	570 (m1) 645 (m2)	1 cycle of 95 °C for 9 min, 40 cycles of 95 °C for 30 s, 52 °C for 30 s and 72 °C for 30 s, and 1 cycle of 72 °C for 5 min [27]
	Intermediate (i1)	F: GTTGGGATTGGGGGAATGCCCG R: TTAATTTAACGCTGTTTGAAG	426	1 cycle of 95 °C for 4 min, 35 cycles of 95 °C for 30 s, 55 °C for 60 s and 72 °C for 30 s, and 1 cycle of 72 °C for 5 min [28,29,30]
	Intermediate (i2)	F: GTTGGGATTGGGGGAATGCCG R: GATCAACGCTCTGATTTGA	432	1 cycle of 95 °C for 4 min, 35 cycles of 95 °C for 30 s, 55 °C for 60 s and 72 °C for 30 s, and 1 cycle of 72 °C for 5 min [28,29,30]
	Deletion 81bp	F: ACTAATATTGGCACACTGGATTTG R: CTCGCTTGATTGGACAGATTG	367–379 (d1) 298 (d2)	1 cycle of 95 °C for 2 min, 35 cycles of 95 °C for 30 s, 53 °C for 60 s and 72 °C for 30 s, and 1 cycle of 72 °C for 5 min [31]
*iceA*	*iceA1*	F: GTGTTTTTAACCAAAGTATC R: CTATAGCCASTYTCTTTGCA	247 (A1)	1 cycle of 95 °C for 2 min, 40 cycles of 94 °C for 30 s, 50 °C for 30 s and 72 °C for 30 s, and 1 cycle of 72 °C for 5 min [23,32]
	*iceA2*	F: GTTGGGTATATCACAATTTAT R: TTRCCCTATTTTCTAGTAGGT	229 (A2)	1 cycle of 95 °C for 2 min, 40 cycles of 94 °C for 30 s, 50 °C for 30 s and 72 °C for 30 s, and 1 cycle of 72 °C for 5 min [23,32]
*babA*	*babA2*	F: AATCCAAAAAGGAGAAAAAGTATGAAA R: TGTTAGTGATTTCGGTGTAGGACA	832	1 cycle of 95 °C for 5 min, 35 cycles of 92 °C for 60 s, 52 °C for 60 s and 72 °C for 60 s, and 1 cycle of 72 °C for 5 min [26,33]

**Table 2 cancers-16-01476-t002:** Characteristics of adult patients with a diagnosis of dyspepsia and *H. pylori* infection in Lima during 2016 and 2017.

		N (%)	Bivariate Analysis			Regression Model ^1^		
			CNAG (*n* = 97, 61.4%) N (%)	IM (*n* = 61, 38.6%) N (%)	*p* Value	RP	95% CI	*p* Value
Sex	Female	111 (70.3)	73 (65.7)	38 (34.3)	0.086	Ref.		
	Male	47 (29.7)	24 (51.1)	23 (48.9)		1.42	0.97–2.11	0.073
	<35 years	19 (12.1)	15 (78.9)	4 (21.1)		Ref.		
Age	35–50 years	46 (29.1)	36 (78.3)	10 (21.7)	0.001	1.03	0.36–2.89	0.951
	>50 years	93 (58.8)	46 (49.5)	47 (50.5)		2.40	0.97–5.88	0.056

^1^ Regression generalized linear model with log link in binomial family. Note: CNAG, Chronic non-atrophic gastritis; IM, Intestinal metaplasia; statistically significant (*p* < 0.05); PR, Prevalence ratio; 95% CI, 95% confidence interval.

**Table 3 cancers-16-01476-t003:** Frequency and association of virulence genotypes with histopathological findings in adult patients diagnosed with dyspepsia and *H. pylori* infection in Lima during 2016 and 2017.

	Genotype	N (%)	Bivariate Analysis			Regression Model ^3^		
			CNAG (*n* = 97, 61.4%) N (%)	IM (*n* = 61, 38.6%) N (%)	*p* Value	RP	95% CI	*p* Value
					1.000	Ref.		
*CagA*	*CagA* positive	158 (100.0)	97 (61.4)	61 (38.6)		-	-	-
					0.206 ^2^			
EPIYA motifs	ABC	108 (68.4)	71 (65.7)	37 (34.3)		Ref.		
	AB	6 (3.8)	4 (83.3)	1 (16.7)		0.48	0.07–2.98	0.436
	AB/ABC	5 (3.2)	3 (60.0)	2 (40.0)		1.16	0.38–3.53	0.784
	ABC/ABCC	28 (17.7)	13 (46.4)	15 (53.6)		1.56	1.02–2.41	0.043
	ABC/ABCC/ABCCC	11 (6.9)	5 (45.5)	6 (54.5)		1.59	0.87–2.91	0.130
					0.120 ^2^			
*vacAs*	*vacAs2*	8 (5.1)	7 (87.5)	1 (12.5)		Ref.		
	*vacAs1*	150 (94.9)	90 (60.0)	60 (40.0)		3.19	0.51–20.34	0.218
					0.036			
*vacAs1*	*vacAs1a*	76 (48.1)	43 (56.6)	33 (43.4)		Ref.		
	*vacAs1a/s1b*	34 (21.5)	28 (82.4)	6 (17.6)		0.40	0.18–0.88	0.022
	*vacAs1a/s1c*	24 (15.2)	13 (54.2)	11 (45.8)		1.05	0.63–1.75	0.834
	*vacAs1a/s1b/s1c*	24 (15.2)	13 (54.2)	11 (54.2)		1.05	0.63–1.75	0.834
					0.009			
*vacAm*	*vacAm2*	29 (18.3)	24 (82.7)	5 (17.3)		Ref.		
	*vacAm1*	129 (81.7)	73 (56.6)	56 (43.4)		2.52	1.11–5.74	0.028
					0.007			
*vacAi*	*vacAi2*	44 (27.9)	35 (79.6)	9 (20.4)		Ref.		
	*vacAi1*	101 (63.9)	53 (52.5)	48 (47.5)		2.32	1.25–4.32	0.008
	*vacAi1/i2*	13 (8.2)	9 (69.2)	4 (30.8)		1.50	0.55–4.11	0.426
					0.785			
*vacAd*	*vacAd2*	46 (29.1)	29 (63.1)	17 (36.9)		Ref.		
	*vacAd1*	112 (70.9)	68 (60.7)	44 (39.3)		1.06	0.68–1.65	0.787
					0.007			
*iceA*	*iceA2*	15 (9.5)	5 (33.3)	10 (66.7)		Ref.		
	*iceA1*	33 (20.9)	16 (48.5)	17 (51.5)		0.77	0.47–1.26	0.301
	*iceA1/iceA2*	110 (69.6)	76 (69.1)	34 (30.9)		0.46	0.29–0.73	0.001
					0.522			
*babA2*	*babA2* negative	75 (47.5)	48 (64.1)	27 (35.9)		Ref.		
	*babA2* positive	83 (52.5)	49 (59.1)	34 (40.9)		1.13	0.76–1.69	0.525
					0.007			
*cagA+/vacAs1m1*	Other genotypes ^1^	33 (20.9)	27 (81.2)	6 (18.2)		Ref.		
	*cagA+/vacAs1m1*	125 (79.1)	70 (56.0)	55 (44.0)		2.42	1.14–5.13	0.021
					0.010			
*cagA+/vacAs1am1*	Other genotypes ^1^	100 (63.3)	69 (69.0)	31 (31.0)		Ref.		
	*cagA+/vacAs1am1*	58 (36.7)	28 (48.3)	30 (51.7)		1.67	1.13–2.45	0.009

^1^ *cagA+* genotypes with allelic variants different from the *vacAs1m1* or *vacAs1am1* genotype. ^2^ Calculated by Fisher’s exact test. ^3^ Regression generalized linear model with log link in binomial family. Note: CNAG, Chronic non-atrophic gastritis; IM, Intestinal metaplasia; statistically significant (*p* < 0.05); PR, Prevalence ratio; 95% CI, 95% confidence interval.

## Data Availability

The data that support this research are compiled in the manuscript prepared by J.G., in compliance with the requirements to obtain the title of Doctor in Epidemiological Research offered by the Universidad Peruana Cayetano Heredia, Lima, Peru, and deposited in the institutional repository with URL https://repositorio.upch.edu.pe/handle/20.500.12866/14822 (accessed on 10 January 2024).

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
