# Peer review of "Helicobacter pylori cagA, vacA, iceA and babA Genotypes from Peruvian Patients with Gastric Intestinal Metaplasia"

_cancers, 2024, doi:10.3390/cancers16081476_

Round 1
Reviewer 1 Report
Comments and Suggestions for Authors
The aim was to determine the association between genes associated with H. pylori virulence factors and the clinical stage of preneoplastic lesion gastric intestinal metaplasia.
1. The study included 158 H. pylori clinical isolates from adult outpatients with dyspepsia and diagnosed with H. pylori infection. a. Why are there no healthy individuals included as a control? b. It Would be interesting if the authors included sample analysis from other clinical stages to verify the changes between a CNAG and AG.
2. In line 136, the authors mentioned that 38.6% of cases were histopathologically diagnosed as IM; where is this percentage shown in Table 1? This table should include in the footnote what CNAG and IM mean.
3. According to the results in Table 3, all the virulence factors evaluated are increased in the CNAG biopsies, and the values in IM seem irrelevant. How do authors explain that?
Comments on the Quality of English LanguageSome grammatical mistakes can be improved.
Author Response
Dear Reviewer
Thank you very much for taking the time to review this manuscript. Please find the detailed responses below and the corresponding revisions and corrections highlighted in the re-submitted files.
Thank you for your consideration of this manuscript.

Reviewer 2 Report
Comments and Suggestions for Authors
Dear Authors,
Thank you very much for submitting your paper to the Surgeries. Below I present point-by-point comments and suggestions regarding your paper:
- Line 17 in the abstract, before using ‘H. pylori’ form, please use a full name – Helicobacter pylori. Similarly, correct it with other abbreviations in the text – firstly a full name should be used before introducing the abbreviation. Further, I would recommend adding ‘Abbreviations’ section at the very end of the manuscript (for instance after the conclusions’ section
- In the abstract, you should provide the number of patients (n=…) who were examined in this study
- In the keywords, I would recommend adding such words as ‘CagA’, ‘VacA’, ‘IceA’, and ‘BabA’.
- Line 57 in the introduction – there are two dots at the end of the sentence – one before and one after the citation. Please correct
- I would recommend adding a separate ‘Limitations of the study’ section, preferably before the discussion.
- It would be beneficial to add a figure with mechanisms of action of chosen virulence factors
Kind regards
Comments on the Quality of English LanguageThere are several grammatical errors that should be corrected before the paper can be processed further. Besides, there are some words that could be replaced by other ones making the paper sounding more scientific. In some sentences, the words used are simply inappropriate and out of context. Please revise
Author Response

(The authors gave the same response as above.)
